# Slack Resources and Export Intensity: New Empirical Evidence from Vietnam

**Thi Thanh Binh Dinh** [1,*], **Thuy Anh Tu** [2] **and Thi Mai Phuong Chu** [2]

1   Faculty of International Economics, Foreign Trade University, Hanoi 100000, Vietnam
2   Journal of International Economics and Management, Foreign Trade University, Hanoi 100000, Vietnam
*   Correspondence: binhdtt@ftu.edu.vn

**Abstract:** The paper explored the effects of financial and human resource slacks on firms' export intensity. Using longitudinal data of Vietnamese firms with a random effect model, the study found that financial slack had an inverted-U shaped effect; meanwhile, human resource slack had no consistently significant effect on firms' export intensity. The empirical findings could have managerial practice implications at the firm level. Too few—or too many—slack resources are bad for export intensity. Therefore, firm managers need to consider the necessary level of redundancy for slack resources to avoid prodigality while doing business. This study contributed new insights to international business and slack literatures. First, our findings contributed to the international business literature. In classic internalization studies, it has generally been believed that firms with access to greater resources are more likely to exploit foreign markets. However, our research showed that, when there were too many slack resources, business activities in firms' foreign markets, such as exports, could be reduced due to complacency among firm managers. Second, the research results contributed to literature on slack resources, providing a new insight on their effects on firms' export intensity. Aside from linear or U-shaped effects, slack resources could have inverse U-shaped relationships with firms' export intensity.

**Keywords:** export intensity; slack resources; financial slack; human resource slack

## 1. Introduction

A great deal of research has been carried out on the factors affecting the export of enterprises. Authors have mainly focused on three sets of resources: firm size, industry, and technology. These important resources are good predictors of a firm's export strategy (Dhanaraj and Beamish 2003). However, a decision maker's personality can affect the ability of a firm to expand its exports, and those export choices can only be implemented if the slack resources of the firm allow that selection to be exercised (Reid 1981).

Slack resources are defined as the excess of a firm's resources, compared with the average resource of the industry (Paeleman et al. 2017), or the difference between the total number of resources and the amount of resources actually being used to guarantee normal production in the enterprise (Cyert and March 1963; Bourgeois 1981). Some scholars have stated that slack resources are reserve resources that enterprises can use against threats or to exploit opportunities in the business environment. Slack also facilitates innovation and permits firms to more safely experiment with new strategies by introducing new products and entering new markets (Moses 1992; Nohria and Gulati 1996). However, some scholars have expressed opposite opinions, arguing that slack resources could negatively affect organizations because they have not been used for value-generation processes, reflecting poor management or even an arbitrary distribution of resources that may generate agency problems (Nohria and Gulati 1996; Tan and Peng 2003).

Although the concept and role of slack resources have been discussed for a long time, studies on the impact of slack resources on firms' export behavior, such as export intensity,

are extremely scarce. Moreover, research results from previous studies in this field remain contradictory. To the best of our knowledge, there have been only three studies related to this field, including Kiss et al. (2017), Lin et al. (2009), and Paeleman et al. (2017). Paeleman et al. (2017) hypothesized that financial slack and human resource slack had U-shaped relationships with firms' export intensity. However, the estimated results of their study of Belgian firms showed that financial slack had no effect on export intensity, while human resource slack had a negative linear effect. Meanwhile, Kiss et al. (2017) found a U-shaped relationship between financial and human resource slack and their impact on the export intensity of small and medium enterprises in France, Italy, and Spain. The study by Lin et al. (2009) on high-technology firms in Taiwan found that high-discretion slack had a U-shaped effect on foreign sales, while low-discretion slack had a positive linear effect on the same. The incompatible findings of previous research suggested the necessity of more studies on this topic, in different economies, with diversified data. In doing so, we could gain a deeper understanding of the role of slack resources in the export activities of firms, especially in the context of an emerging economy like Vietnam, with exports being one of the main drivers of economic development.

This study used panel data on 12,497 exporting firms operating in Vietnam, for five years, from 2015 to 2019. Vietnam was a suitable choice for a study about an emerging economy. Vietnam's shift from a centrally-planned to a market-based economy has transformed the country from one of the poorest in the world into a lower middle-income country. Vietnam is one of the most dynamic emerging countries in East Asia region (Gupta et al. 2014). One of the characteristics of emerging economies is that they become more integrated with the global economy through various forms of international business—in which firms' export activities play an important role (Aulakh et al. 2017; D. A. Singh 2009).

To estimate the effects of slack resources on firms' export intensity, we applied the random effect model. The estimated results showed that the effect of slack resources, particularly financial slack, on firms' export intensity was inverse U-shaped. Having too little or too much financial slack is both bad for export intensity. Too little financial slack will make exporting activities difficult because of the many costs involved. However, too much financial slack also leads to complacency in firm managers, thinking that the business is doing very well and thereby reducing the incentive to find new markets to increase export intensity.

This study provided new insight on the impacts of slack resources on firms' export intensity. Our research contributed to both international business literature and slack literature by providing a better understanding of whether or not different types of organizational slack effected firms' export strategies. The empirical results could be particularly important for managers of firms that are involved in export or tend to enter into exporting.

The structure of this paper was organized as follows. In Section 2, we reviewed theories on the effects of slack resources on the export intensity of firms and set up hypotheses of the study. In Section 3, we presented our research methodology. In Section 4, we analyzed empirical results. The final section was devoted to discussion and conclusion.

## 2. Theory and Hypotheses

### 2.1. Slack Resources

Although the importance of slack resources and their definition goes back to the works of Cyert and March (1963), it was Bourgeois (1981) who established the guidelines for the current development of the concept and empirical analysis.

In fact, slack resources have been defined diversely in many studies. Cyert and March (1963) defined slack as "the difference between total resources and total necessary payments". Bourgeois (1981) followed this definition and added that "organizational slack is that cushion of actual or potential resources which allows an organization to adapt successfully to internal pressures for adjustment or to external pressures for changes in policy, as well as to initiate changes in strategy with respect to the external environment". Based on the definition of Bourgeois (1981), Sharfman et al. (1988) emphasized two characteristics of

slack. First, slack must be visible to the manager and employable in the future. Second, different types of slack give managers dissimilar degrees of discretion in attempting to protect their firms from internal and external pressures. Nohria and Gulati (1996) defined slack as "a pool of resources in an organization that is in excess of the minimum necessary to produce a given level of organizational output". The authors stated that slack resources included excess inputs such as redundant employees, unused capacity, and unnecessary capital expenditures. George (2005) defined slack as "potentially utilizable resources that can be diverted or redeployed for the achievement of organizational goals". This study adopted the definition by Nohria and Gulati (1996) for three reasons. First, it addressed the fact that slack can act as a cushion, protecting firms from external pressures. Thus, we expected that this type of cushion could affect a firm's internationalization, such as exporting activities. Second, this definition emphasized that slack allows firms to initiate change, potentially fostering a firm's exporting activities. Third, as different forms of slack give managers different degrees of flexibility to reduce pressures, consequently, it was expected that higher or lower discretion slack could have different effects on the exporting activities of a firm.

Along with the diverse definitions, slack resources have been classified into various types by researchers. Bourgeois and Singh (1983) classified slack into three types based on the degree of accessibility: (1) available slack is a resource with high discretionary power that can be used flexibly and is not committed to a specific area of consumption in a firm, (2) recoverable slack is a cost greater than the cost required for firm operations which can be recovered from a firm's reorganization although it has already been absorbed as a cost, (3) potential slack refers to future resources generated through debt borrowing. Most studies use the current ratio (Agusti-Perez et al. 2020; Ahuja 2000; Bergh 1997; Bromiley 1991; Chang and Singh 1999; B.-N. Kim et al. 2017; Martinez and Artz 2006) and quick ratio (Combs and Ketchen 1999; Davis and Stout 1992; Geiger and Cashen 2002; Palmer and Wiseman 1999) as measurements for available slack; the ratio of R&D investment to sales, the advertising expense to sales or human resources as the proxies for recoverable slack (B.-N. Kim et al. 2017; Kiss et al. 2017); and the debt/equity, debt/sales and debt/assets ratios as the measurement for potential slack (Bergh 1997; Chang and Singh 1999; Graves and Waddock 1994; Zahra 1996).

Organizational slack has also been subclassified into high-discretion slack and low-discretion slack based on managerial discretion in the deployment of resources (Sharfman et al. 1988). The authors suggested that slack resources were anchored along a continuum of managerial discretion, wherein low-discretion slack provided less flexibility to managers and their strategic options. While more discretionary slack resources can be used in a wide variety of situations, less discretionary slack resources can be used in only a few specific situations (Lin et al. 2009). Examples of high-discretion slack include cash (George 2005), the sum of cash and marketable securities (J. V. Singh 1986), current ratio (Agusti-Perez et al. 2020; Ahuja 2000; Bergh 1997; Bromiley 1991; Chang and Singh 1999; B.-N. Kim et al. 2017; Martinez and Artz 2006; Lin et al. 2009), and quick ratio (Combs and Ketchen 1999; Davis and Stout 1992; Geiger and Cashen 2002; Palmer and Wiseman 1999). Examples of low-discretion slack include debt (Illinitch and Zeithaml 1995), equity/debt ratio (Bourgeois and Singh 1983; Combs and Ketchen 1999; Lin et al. 2009), debt/equity ratio (Bergh 1997; Bergh and Lawless 1998; Bromiley 1991; George 2005; Martinez and Artz 2006), and debt/assets ratio (Graves and Waddock 1994).

J. V. Singh (1986) measured slack using two approaches: unabsorbed and absorbed slack. Absorbed slack refers to excess overhead in organizations that is not easy to redeploy. Unabsorbed slack, which corresponds to currently uncommitted resources, is more easily redeployed elsewhere, allowing for greater managerial discretion. While both types of slack are subject to managerial discretion, there are significant differences in how quickly they are available for deployment and their potential impact on the performance of the firm (Wefald et al. 2010). Examples of absorbed slack include general and administrative expenses and human resources (Nguyen et al. 2019; Wefald et al. 2010). Examples of unabsorbed

slack include cash (George 2005), depreciation funds (Tan and Peng 2003), current ratio (Agusti-Perez et al. 2020; Ahuja 2000; Bergh 1997; Bromiley 1991; Chang and Singh 1999; B.-N. Kim et al. 2017; Martinez and Artz 2006; Lin et al. 2009), and quick ratio (Combs and Ketchen 1999; Davis and Stout 1992; Geiger and Cashen 2002; Palmer and Wiseman 1999).

Regarding the role of slack resources in organizations, many empirical studies have been carried out. Researchers have focused on organizational slack, in different forms, as a predictor of organizational performance (George 2005; B.-N. Kim et al. 2017; Tan and Peng 2003; Wefald et al. 2010), organizational innovation (B.-N. Kim et al. 2017; Kiss et al. 2017; Nohria and Gulati 1996), R&D investment (P. M. Kim et al. 2008), risk-taking (Wiseman and Bromiley 1996), internationalization (Lin et al. 2009; Nguyen et al. 2019), and exporting behavior (Kiss et al. 2017; Paeleman et al. 2017). However, the results were not conclusive and left an opportunity to deepen the research in this field. Among possible explanations for the divergence of results is the large array of approaches, theories, and measures that have been applied for both slack and organizational outcome.

From a general perspective, two theoretical approaches have dominated studies: the organization theory and the agency theory. According to the organization theory, organizational slack has a positive effect on organizations. Cyert and March (1963) suggested that slack may buffer against fluctuation in different environments, thus absorbing environmental shocks. Accordingly, organizational slack serves as a resource for conflict resolution, a buffering mechanism in the workflow process, and a facilitator of strategic or creative behavior within the organization (Lin et al. 2009). Moreover, slack facilitates innovation, and permits firms to more safely experiment with new strategies by introducing new products and entering new markets (Moses 1992; Nohria and Gulati 1996).

Opposite to the organization theory, the agency theory posits that slack resources could negatively affect organizations because these resources have not been used for value-generation process, reflecting poor management or even an arbitrary distribution of resources that may generate agency problems (Nohria and Gulati 1996; Tan and Peng 2003). In this respect, maintaining slack can be good for organizations, but only when managers act as agents (Tan and Peng 2003). When the interests of managers are not aligned with those of principals, managers may use slack resources inefficiently, becoming a source of agency problems.

However, many studies have shown a curvilinear relationship between slack and organization performance. In other words, there is a contradictory relationship that is both positive and negative, with performance depending on the amount of organizational slack (Bourgeois 1981; P. M. Kim et al. 2008; Nohria and Gulati 1996; Tan and Peng 2003). Bourgeois (1981) pointed out that a firm would normally pursue sufficient slack resources for unexpected situations; however, when slack resources exceed a certain amount, it can negatively impact a firm's performance due to the interests of managers. The empirical works reported an inverted U-shaped relationship between slack resources, in different measurements, and innovation (Nohria and Gulati 1996), firm performance (George 2005; Tan and Peng 2003), firm globalization (Tseng et al. 2007), and R&D investment (P. M. Kim et al. 2008).

### 2.2. Slack Resources and Firms' Export Intensity

From the organization theory, slack resources can provide firms with a means of crossing borders, allowing them to compete in international markets with fewer constraints (Cyert and March 1963; Lin et al. 2009; Tseng et al. 2007). Therefore, slack resources are expected to promote export intensity of enterprises. However, the agency theory notes downsides of having too much slack. Excessive slack resources can also insulate firms from external pressure, thereby reducing motivations to adapt to environmental pressures and engage in contingency projects (P. M. Kim et al. 2008; Lin et al. 2009; Nohria and Gulati 1996). From this point of view, slack resources can be expected to reduce the export intensity of firms.

Empirical studies on the effects of slack resources on firms' export intensity also have conflicting results. Research results by Paeleman et al. (2017) on the effects of financial and human resource slack on Belgian firms' export behavior showed that financial slack had no effect on export intensity, but human resource slack had a negative linear effect on firms' export intensity. Meanwhile, Kiss et al. (2017) found a U-shaped relationship between financial and human resource slack on the export intensity of small and medium enterprises in France, Italy, and Spain. The study of Lin et al. (2009) found a U-shaped effect of high-discretion slack, proxied by current ratio, and a positive linear effect of low-discretion slack, measured by equity/debt ratio, on foreign sales of high-technology firms in Taiwan.

In this study, we utilized the research methods of Paeleman et al. (2017) to study the effects of slack resources on the export intensity of enterprises in Vietnam. As such, the study focused on financial and human resource slack, which "lie at opposing ends of a continuum" (Kiss et al. 2017), representing the redeployment ability of slack resources. While financial slack is classified as high-discretion slack, human resource slack is classified as low-discretion slack.

Financial slack represents excess uncommitted financial resources that are highly flexible and can be applied to a wide range of activities (Mishina et al. 2004). Financial slack helps firm managers be more confident in their decisions to expand and develop their businesses, such as participating in export markets or increasing export activities. Additionally, there are many costs involved in exporting activities, such as the cost of finding market information, the cost of training human resources, the cost of establishing distribution networks in a new market, or the cost of developing and manufacturing products in line with the needs of new markets. Financial slack helps managers to cover such additional costs so they can confidently engage in and increase export activities abroad (Paeleman et al. 2017). However, too much financial slack will make business managers feel complacent, as the business is doing very well, thereby reducing the incentive to find new markets, to increase export intensity, and to generate more profits for the firm (Bourgeois 1981; P. M. Kim et al. 2008; Paeleman et al. 2017). In summary, we suggested that:

**Hypothesis 1.** *The relationship between financial slack and export intensity is inverse U-shaped.*

Human resources refers to the number of employees in excess of a firm's operational needs (Bourgeois 1981; Mishina et al. 2004). Human resource slack is more closely committed to the operations of firms and has an increased degree of specificity and path dependence associated with it. It takes significant time and financial resources to train an employee to become knowledgeable about business operations. Therefore, human resource slack makes it easier for enterprises to arrange personnel for new tasks arising from engagement in export activities (Paeleman et al. 2017). This can reduce anxiety and concern over foreign market risks, as firms can afford to make mistakes and experiment with new internationalized strategies, such as exploring new foreign markets or increasing export activities to present partners. However, having too much human resource slack can also drive firm managers to exploit human resource less efficiently, therefore decreasing the intensity of exporting (Lin et al. 2009; Paeleman et al. 2017). Thus, this study hypothesized that:

**Hypothesis 2.** *The relationship between human resource slack and export intensity is inverse U-shaped.*

With these two hypotheses, our study intended to find new aspects of the influence of slack resources on export intensity of firms that differed from the research results of previous studies. Aside from the linear or U-shaped effects of slack resources on export intensity that were found in the previous studies, we hypothesized that slack resources could have an inverse U-shaped relationship with firms' export intensity. Our findings

could reveal new insights into the role of slack resources in export activities of firms and contribute to the literature on international business and slack.

## 3. Methods

### 3.1. Sample

The sample data used in this study were obtained from surveys of enterprises operating in Vietnam, conducted by the General Statistics Office of Vietnam, from 2015 to 2019. These were comprehensive surveys, covering all state enterprises, non-state enterprises that with 10 or more employees, 20% of sampled non-state enterprises with fewer than 10 employees, and all foreign enterprises across 64 provinces and cities in Vietnam. The longitudinal capacity of the dataset, i.e., that each firm was identified through a unique tax code, allowed a firm to be followed over time. After filtering out samples not following normal standards, such as those with negative assets or a negative number of employees, we constrained for a suitable range, according to the entire variable list, to eliminate outliers. Finally, the study used a panel data of 12,485 exporting firms for five years from 2015 to 2019.

### 3.2. Variables and Measures

*Dependent variable*. The export intensity was the dependent variable, measured as the ratio of a firm's exporting sales to total sales in a given year (Kiss et al. 2017; Paeleman et al. 2017).

*Independent variables*. The independent variables measured financial slack and human resource slack. Adopting the definition of slack by Nohria and Gulati (1996) and the methods suggested by Paeleman et al. (2017), based on the available data of Vietnam's firms, financial slack was measured as the difference between the revenue-to-asset ratio of firm $i$ and the average value of this ratio in the same 2-digit industry $j$. Human resource slack was measured as the difference between the ratio of full-time employees to assets of firm $i$ and the average value of this ratio in the same 2-digit industry $j$.

*Control variables*. This study included five control variables for firm characteristics: firm size, firm age, firm location, firm ownership, and firm industry. *Firm size* as a significant factor affecting export intensity (Bonaccorsi 1992; Verwaal and Donkers 2002). This factor represented a strong capability and an abundance of resources to cope with the difficulties of exploring new foreign markets (Dunne et al. 1989). Firm size was measured as logarithmic values of assets, labors, and sales in a given year. Regarding firm location, Meyer (1998) stated that, in transition economies, locating industrial zones, or export processing zones, firms benefit from good infrastructure conditions, supporting services relating to administrative procedures, priority policies related to profit tax, import duties, and land use fees. These conditions create many advantages for firms to implement new internationalized strategies, such as export intensity. *Firm location*, herein, was a dummy variable that took the value of one, if firms were located in an industrial zone or an export processing zone. Lin et al. (2009) suggested that older firms had, relatively, more international market commitment and organizational resources, which affected their internationalization, e.g., export activities. *Firm age* was measured as a logarithm of firm operating years. Firm ownership was included as a control variable. Many studies showed that firm ownership affected the export activities of enterprises in general, and the export intensity of enterprises in particular (Javalgi et al. 2000; Kiss et al. 2017; Zhao and Zou 2002). *Firm ownership* was classified into three categories: state-owned enterprises, foreign-owned enterprises, and other kinds of firms, such as private firms and household businesses. The variable *state-owned enterprise* (SOE) was a dummy variable that equaled one if firms had state capital greater than 50%. The variable *foreign-owned enterprise* (FOE) was a dummy variable that equaled one if firms had foreign capital. The study also controlled for firm industry. Many studies showed that the industry characteristics of enterprises had an effect on the intensity of exports (Reis and Forte 2016). Firm industry was classified into three categories: manufacturing, agro-forestry-fishery, and other kinds of industries. The

variable manufacturing was a dummy variable that equaled one if the firm was operating in the manufacturing sector. The variable agro-forestry-fishery was a dummy variable that equaled one if the firm was operating in that sector.

### 3.3. Model Specification

To investigate the effects of financial and human resource slack on export intensity, the study used the statistical software Stata to analyze the panel data on Vietnamese firms. Since the study used panel data of firms, we applied the Dickey–Fuller test to check for the stationarity of the variables (unit root test). The results showed that all the variables of the model were stationary. In this case, three econometric models—the pooled ordinary least square model, random-effect model, and fixed-effect model—were potential suitable choices. As the model included some dummy independent variables whose values did not change over time, the random-effect model and pooled OLS model were also suitable (Wooldridge 2019). The Breusch and Pagan Lagrange multiplier test for random effects confirmed the panel effect, meaning that there were significant differences across units. Therefore, the random effect model was chosen. We applied the Breusch–Pagan test to check for heteroskedasticity and the Wald test for serial correlation. The tests confirmed that the models had heteroskedasticity, but not serial correlation. To deal with heteroskedasticity, we applied the robust standard errors method for the random effect model. (See Supplementary File S1 for the tests)

Table 1 presents the descriptive statistics and correlation matrix for the variables. The correlation matrix suggested that there was no multicollinearity in the models, as the absolute values of correlation coefficients of independent variables were all smaller than 0.8.

**Table 1.** Descriptive statistics and correlations.

| | Variable | Mean | Std. Dev. | 1 | 2 | 3 | 4 | 5 | 6 | 7 | 8 | 9 | 10 | 11 | 12 |
|---|---|---|---|---|---|---|---|---|---|---|---|---|---|---|---|
| 1 | Export intensity | 0.733 | 27.779 | 1.000 | | | | | | | | | | | |
| 2 | Financial slack | 2.167 | 319.820 | −0.001 | 1.000 | | | | | | | | | | |
| 3 | HR slack | 0.005 | 0.381 | 0.005 | 0.282 | 1.000 | | | | | | | | | |
| 4 | Lnturnover | 8.373 | 2.119 | −0.067 | 0.027 | −0.033 | 1.000 | | | | | | | | |
| 5 | Lnasset | 8.815 | 1.692 | −0.021 | −0.055 | −0.065 | 0.828 | 1.000 | | | | | | | |
| 6 | Lnlabor | 2.285 | 1.332 | −0.028 | −0.007 | 0.010 | 0.707 | 0.661 | 1.000 | | | | | | |
| 7 | Lnage | 2.427 | 0.415 | −0.010 | −0.014 | −0.015 | 0.267 | 0.298 | 0.290 | 1.000 | | | | | |
| 8 | Firm location | 0.037 | 0.190 | −0.007 | −0.015 | −0.012 | 0.243 | 0.290 | 0.256 | −0.003 | 1 | | | | |
| 9 | SOE | 0.009 | 0.094 | −0.005 | −0.001 | −0.004 | 0.142 | 0.174 | 0.133 | 0.198 | −0.074 | 1 | | | |
| 10 | FOE | 0.036 | 0.185 | −0.006 | −0.011 | −0.001 | 0.105 | 0.084 | 0.127 | −0.108 | 0.415 | −0.158 | 1 | | |
| 11 | Manufacturing | 0.015 | 0.120 | 0.006 | −0.003 | −0.002 | −0.009 | 0.046 | 0.04 | 0.057 | −0.034 | 0.146 | −0.002 | 1 | |
| 12 | Agro-forestry-fishery | 0.182 | 0.385 | −0.012 | −0.022 | −0.002 | 0.194 | 0.19 | 0.417 | 0.131 | 0.411 | −0.067 | 0.159 | −0.11 | 1 |

## 4. Results

Table 2 reports the results of the robust random effect regression. We set up four econometric models to see if the estimated results were consistent. The first model included only the control variables, serving as a baseline from which the analysis proceeded. In Model 2, we included the control and financial slack effects. Model 3 introduced the control and human resource slack effects. Model 4 included all the control, financial slack, and human resource slack effects.

**Table 2.** Robust random effect regression results for export intensity.

| Variables | Model 1 | | Model 2 | | Model 3 | | Model 4 | |
|---|---|---|---|---|---|---|---|---|
| | Coef. | SE | Coef. | SE | Coef. | SE | Coef. | SE |
| Financial slack | | | 0.014 ** | 0.006 | | | 0.015 ** | 0.007 |
| Financial slack squared | | | $-2.42 \times 10^{-6}$ ** | $1.06 \times 10^{-6}$ | | | $-2.62 \times 10^{-6}$ ** | $1.20 \times 10^{-6}$ |
| Human resourceslack | | | | | 2.581 * | 1.329 | −1.126 | 1.333 |
| Human resourceslack squared | | | | | −0.104 * | 0.060 | 0.053 | 0.058 |
| Lnturnover | −1.297 *** | 0.395 | −1.341 *** | 0.411 | −1.302 *** | 0.397 | −1.343 *** | 0.413 |
| Lnasset | 0.791 *** | 0.301 | 0.842 *** | 0.319 | 0.811 *** | 0.308 | 0.839 *** | 0.318 |
| Lnlabor | 0.314 *** | 0.117 | 0.315 *** | 0.117 | 0.301 *** | 0.115 | 0.320 *** | 0.121 |
| Lnage | −0.003 | 0.163 | −0.0059 | 0.164 | −0.002 | 0.163 | −0.006 | 0.164 |
| Firm location | 0.059 | 0.104 | 0.0046 | 0.105 | 0.054 | 0.105 | 0.046 | 0.105 |
| SOE | −0.099 | 0.148 | −0.113 | 0.151 | −0.103 | 0.149 | −0.113 | 0.151 |
| FOE | 0.045 | 0.095 | 0.056 | 0.095 | 0.048 | 0.095 | 0.056 | 0.095 |
| Manufacturing | −0.708 | 1.105 | −0.743 | 1.109 | −0.707 | 1.105 | −0.747 | 1.111 |
| Agro-forestry-fishery | −0.360 | 0.196 | −0.346 | 0.192 | −0.351 | 0.194 | −0.347 | 0.193 |
| Wald-chi square | 29.85 *** | | 29.81 *** | | 41.86 *** | | 33.02 *** | |
| Number of firms | 12497 | | 12497 | | 12497 | | 12497 | |

* Significant at 10% level. ** Significant at 5% level. *** Significant at 1% level.

The results of the baseline model were consistent with the other models. Among the control variables, only three variables representing firm size had a statistically significant effect on the export intensity of firms, including business turnover, assets, and number of employees. However, business turnover had a negative effect, while assets and number of employees had a positive effect, on the export intensity of firms.

Concerning slacks, Hypothesis 1 predicted that the relationship between financial slack and export intensity was inverse U-shaped. As shown in Model 2, the coefficients for financial slack were significant and positive ($B = 0.014$, $p < 0.05$). Meanwhile, the coefficients for financial slack squared were significant and negative ($B = -2.42 \times 10^{-6}$, $p < 0.05$). In Model 4, as in Model 2, the coefficients for financial slack were significant and positive ($B = 0.015$, $p < 0.05$). Meanwhile, the coefficients for financial slack squared were significant and negative ($B = -2.42 \times 10^{-6}$, $p < 0.05$). These statistically significant results indicated an inverse U-shaped effect of financial slack on firms' export intensity. Thus, Hypothesis 1 was supported.

Alternatively, Hypothesis 2 predicted that the relationship between human resource slack and export intensity was inverse U-shaped. The results in Model 3 showed that the coefficient for human resource slack was significant and positive ($B = 2.581$, $p < 0.1$) while the coefficient for human resource slack squared was significant and negative ($B = -0.104$, $p < 0.1$). However, the coefficients for both human resource slack ($B = -1.126$, $p > 0.1$) and human resource slack squared ($B = 0.053$, $p > 0.1$) were not significant in Model 4, and demonstrated opposite signs, compared to those in Model 3. These results indicated that the estimated results of human resource slack effect on export intensity were not consistent. These results were not in line with our expectations. Thus, Hypothesis 2 was not supported.

## 5. Discussion and Conclusions

This article studied the influence of different types of slack resources on the export intensity of firms. The results supported the hypothesis of an inverted U-shaped relationship between financial slack and firm export intensity. This result was different from the result of (Paeleman et al. 2017), which confirmed that financial slack had no statistically significant effect on the export intensity of firms. This result was also opposed to the research result of Kiss et al. (2017) which found financial slack to have a U-shaped effect on the export intensity of small and medium enterprises. As financial slack is considered high-discretion slack (Bergh 1997; George 2005; Tseng et al. 2007), our findings also differed from the study

of Lin et al. (2009) that revealed a U-shaped effect of high-discretion slack on foreign sales of high-technology firms.

Based on the work of Nohria and Gulati (1996), we proposed two basic mechanisms to explain the inverted U-shaped relationship between financial slack and firm export intensity. The first mechanism was the effect of financial slack on the initiation of export activities. The second mechanism was the effect of financial slack on the maintenance and development of export activities. Too little financial slack will can make it difficult for firms to start exporting activities because of the many costs involved in starting exports, such as the costs of finding market information, training staff, establishing distribution networks in a new market, and developing and manufacturing products to satisfy the demands of the new market. However, too much financial slack also makes firm managers feel complacent, thinking that their business is doing very well, thereby reducing the incentive to find new markets to increase export intensity and generate more profits for the firm. Taken together, these arguments suggest that the proper way to think about the relationship between financial slack and export intensity is to view it as having an inverse U-shape. Given the sample we used in this paper, the turning point of the financial slack equaled 2862.595 million Vietnamese dong, equivalent to about $122,072 USD.[1]

The study had no stable results relating to the effects of human resource slack on the export intensity of firms. When there was no financial slack factor in the model, human resource slack had an inverted U-shaped effect on export intensity. However, when the financial slack factor was added to the model, the human resource slack no longer had an effect on firm export intensity. The inconsistent effects of human resource slack somehow reflected the low-discretion characteristics of this factor. As mentioned above, human resource slack is classified as low-discretion slack because the value of human resources is often more context-dependent and connected to existing organizational routines rather than the value of financial capital (Mishina et al. 2004). Given this strong connection to existing organizational routines, the knowledge and skills of human resources are difficult to transfer and exploit between firm branches across countries, as compared to financial resources. These results were not in line with the findings of Paeleman et al. (2017) which revealed a negative linear effect of human resource slack on firms' export intensity. Our findings were also not consistent with the empirical result of Kiss et al. (2017) indicating an U-shaped relationship between human resource slack and the export intensity of small and medium enterprises.

With the estimated results, the paper contributed to two interrelated streams of literature. First, our findings contributed to international business literature. In classic internationalization studies, it has generally been believed that firms with access to more value resources are more likely to exploit foreign markets (Hitt et al. 2006; Rugman 1986). However, our research showed that, when slack resources were too high, business activities in foreign markets of firms, such as exports, could be reduced due to complacency on the part of the firm managers. Second, the research results contributed to slack literature, providing a new insight into the effects of slack resources on firms' export intensity. Aside from linear or U-shaped effects, slack resources could have inverse U-shaped relationships with firms' export intensity. The different findings of our studies suggested that more research in this field needs to be carried out, in different economies with diversified data of firms. Thus, we could gain a better understanding of the impacts of slack resources on firms' export activities.

The empirical findings of the study had some managerial practice implications. The research showed the importance of financial slack to the export activities of enterprises. Slack resources give firms an ownership advantage toward control of their domestic operations and expansion of overseas operations. However, overly-abundant slack resources tend to reduce the intensity of a firms' export, due to the complacent attitude of firm managers. Therefore, managers should frequently remind themselves to control complacent and overly optimistic attitudes. On the other hand, managers need to understand the effects of different types of slack resources. Abundant high-discretion slack, such as financial

slack, can help firms expand their exports. However, once the high-discretion slack rises to a point (turning point), it can have a negative effect on a firm's export performance. Meanwhile, abundant low-discretion slack, such as human resource slack, had no effect on the export activities of enterprises. Therefore, firm managers need to consider the necessary level of redundancy of this type of resource to avoid prodigality in doing business.

The study had some limitations. First, our sample only included enterprises operating in Vietnam, an emerging economy. Therefore, the generalizability of our results is limited, because we did not consider samples from other economies such as developed and newly-industrialized economies. Differences in these types of economies can lead to different effects of slack resources on the intensity of a firm's exports. Second, the study only focused on examining the effects of financial slack, representing high-discretion slack, and human resource slack, representing low-discretion slack. It must be recognized that organizational slack can be represented by many other factors, in addition to the two above. Third, the study did not examine differences in the effects of slack resources on the export intensity in different firm groups, e.g., by industry, ownership type, or innovation level.

These limitations could present opportunities for future research. First, future studies need to study the effects of slack resources on the export intensity of enterprises in different economies, in order to generate comparative data. Second, future research could use many different measures for slack resources, such as current ratio, quick ratio, and retained earnings, to measure high-discretion slack and debt, equity/debt ratio, and debt/asset ratio as proxies for low-discretion slack. Third, future research should consider the impact of other factors, such as firm characteristics, on the relationship between slack resources and export intensity, by adding to the model interaction variables or studying sub-samples classified along with firm characteristics.

In conclusion, we would like to reiterate that recognizing that slack resources, specifically financial slack, have an inverse U-shaped effect on firm export intensity not only represents an important theoretical contribution, but could have great practical significance. To our knowledge, this was the first study on the effects of slack resources on firm export intensity with a large longitudinal sample of firms in an emerging country. We hope that this study has provided a better understanding of the manner in which organizational slack impacts the export behavior of firms.

**Supplementary Materials:** The following are available online at https://www.mdpi.com/article/10.3390/economies11020068/s1, File S1: The Results of the Econometric Test.

**Author Contributions:** Conceptualization, T.T.B.D. and T.A.T.; Methodology, T.T.B.D.; Software, T.T.B.D.; Validation, T.T.B.D., T.A.T. and T.M.P.C.; Formal analysis, T.T.B.D.; Investigation, T.T.B.D.; Resources, T.T.B.D.; Data curation, T.T.B.D.; Writing—original draft, T.T.B.D.; Writing—review and editing, T.T.B.D.; Visualization, T.T.B.D.; Supervision, T.T.B.D.; Project administration, T.T.B.D.; Funding acquisition, T.A.T. All authors have read and agreed to the published version of the manuscript.

**Funding:** This research was funded by Ministry of Education and Training of Vietnam, grant number B2021-NTH-04. And The APC was funded by Ministry of Education and Training of Vietnam.

**Informed Consent Statement:** Not applicable.

**Data Availability Statement:** The data are not publicly available due to the restrictions applied to the availability of these data. Data were obtained from General Statistics Office of Vietnam (GSO) and are available from the authors with the permission of GSO once this research is accepted for publication.

**Conflicts of Interest:** The authors declare no conflict of interest. The funders had no role in the design of the study; in the collection, analyses, or interpretation of data; in the writing of the manuscript; or in the decision to publish the results.

## Note

[1] The turning point is calculated by setting the first derivative of the dependent variable with respect to the financial slack variable to zero.

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
