# Peer review of "Slack Resources and Export Intensity: New Empirical Evidence from Vietnam"

_economies, doi:10.3390/economies11020068_

Round 1
Reviewer 1 Report
This is a very interesting study if could estimate the cut-off value of both slacks in relation to export intensity. Therefore I am suggesting minor revisions if could bring forward some discussion and results on the above lines.
Author Response
From the authors of study entitled “Slack resources and export intensity: A study of an emerging economy” submitted to “Economies”; Manuscript ID: economies-2060782
To: Reviewer 1
Subject: Responses to reviewer’s comments
Dear Reviewer,We would like to thank you for your comments to our study. Hereby, we would like to send you our responses to your comments.
Your comments: This is a very interesting study if could estimate the cut-off value of both slacks in relation to export intensity. Therefore, I am suggesting minor revisions if could bring forward some discussion and results on the above lines.
Our responses: We have decided to not estimate the cut-off value of both slacks because the main purpose of this study is to find out how slack resources effect export intensity of firms. We are more interested in predicting the trend of the effect instead of looking at a specific value. However, we have revised some parts to make the paper better that you can see at the revised version of our paper.
We understand that some of our views could be different from yours. We therefore would like to ask for your consideration.
We sincerely thank you for your time!
Best regards,
The authors
Reviewer 2 Report
This objective of the paper is to explore the effects of financial and human resource slacks on export intensity of firms in Vietnam. The paper tackles an interesting topic. Though the paper is fairly well written, there are a few grammatical mistakes.
The introduction and theory and hypotheses sections are good. Though the authors have not conducted a separate literature review, the paper also demonstrates an adequate understanding of the relevant literature in the field and cites an appropriate range of literature sources. However, there are several papers that are cited in the body of the paper are not listed under references.
Authors have explained the research methodology well in the methods section. The estimation method used to estimate the specified model is also appropriate. However, in Line 218, authors have stated that the study is based on data of 62,485 exporting firms, the figure represents to size of the panel data set. The actual number of firms should be 12,497 not 62,485.
However, the results section needs some improvement. Authors have presented only a correlation matrix and random effects regression results in Tables 1 and 2. Some additional results such as the results of panel unit-root tests, panel cointegration tests, and Hausman test could have been presented. Authors have not also compared the findings of this study with that of previous studies.
The conclusions of the paper adequately tie together with the other elements of the paper. The paper, to some extent, has expressed its case, measured against the technical language of the field and the expected knowledge of the journal's readership.
Author Response
From the authors of study entitled “Slack resources and export intensity: A study of an emerging economy” submitted to “Economies”; Manuscript ID: economies-2060782
To: Reviewer 2
Subject: Responses to reviewer’s comments
Dear Reviewer,We would like to thank you for your comments to our study. Hereby, we would like to send you our responses to your comments.
Your comments: This objective of the paper is to explore the effects of financial and human resource slacks on export intensity of firms in Vietnam. The paper tackles an interesting topic. Though the paper is fairly well written, there are a few grammatical mistakes.
Our responses: We have corrected grammatical mistakes.
Your comments: The introduction and theory and hypotheses sections are good. Though the authors have not conducted a separate literature review, the paper also demonstrates an adequate understanding of the relevant literature in the field and cites an appropriate range of literature sources. However, there are several papers that are cited in the body of the paper are not listed under references.
Our responses: We have checked again and found that all the papers that are cited in the body of the paper are listed under references. We have used a citation software to insert bibliography automatically.
Your comments: Authors have explained the research methodology well in the methods section. The estimation method used to estimate the specified model is also appropriate. However, in Line 218, authors have stated that the study is based on data of 62,485 exporting firms, the figure represents to size of the panel data set. The actual number of firms should be 12,497 not 62,485.
Our responses: We have corrected it.
Your comments: However, the results section needs some improvement. Authors have presented only a correlation matrix and random effects regression results in Tables 1 and 2. Some additional results such as the results of panel unit-root tests, panel cointegration tests, and Hausman test could have been presented. Authors have not also compared the findings of this study with that of previous studies.
Our responses: We have done the The Breusch and Pagan Lagrange multiplier test for random effects; the Breusch – Pagan test to check for heteroskedasticity; Wald test for serial correlation. In our opinion, unit-root test is neccesary only in the case you want to do forecasting (In fact, there are 2 schools of thought that it is neccessay and it is not neccessay). We have not done Hausman test as we mentioned in the paper: “Since the model includes some dummy independent variables, two suitable models are random-effect model (RE) and pooled OLS model (POLS)”. Hausman test is used to select either REM or FEM.
Your comments: The conclusions of the paper adequately tie together with the other elements of the paper. The paper, to some extent, has expressed its case, measured against the technical language of the field and the expected knowledge of the journal's readership.
Our responses: Thank you!
We have also revised some parts to make the paper better that you can see at the revised version of our paper.
We understand that some of our views could be different from yours. We therefore would like to ask for your consideration.
We sincerely thank you for your time!
Best regards,
The authors
Reviewer 3 Report
REFEREE'S REPORT ON
" Slack resources and export intensity: A study of an emerging economy "
COMMENTS: The authors studied “the relationship between the slack resources and export intensity” in Vietnam. This paper needs improvement as has been listed below:
1. Abstract
Ø Explaining the method and variable in one sentence would be sufficient.
Ø The policy proposal of the study's originality and importance should be written in this section.
Ø Findings should be explained with a policy recommendation in this section
2. Introduction
Ø The main motivation of the study should be explained correctly in this section.
Ø The general plan of the study should be explained in this section.
Ø Theoretical explanations regarding the relationship between the slack resources and export intensity are insufficient. A lot of and unnecessary explanations have been made about "Slack resources". Instead, more information should have been written about the relationship between the slack resources and export intensity and its theoretical justifications.
Ø The study's importance, purpose and theoretical framework should be discussed in detail. For example, why the example of Vietnam was chosen even though it was written as emerging economy in the title? Why Vietnam was chosen and its strong relationship with the subject must be written.
Ø The theoretical justification for the hypotheses should be discussed.
Ø Typo on Line 18-22 should be fixed
3. Literature
Ø Studies in the literature on the relevant subject should be written and discussed under the LITERATURE TITLE.
Ø It is also very important to approach studies critically. More recent and current studies should be added to these citations.
Ø The difference of the study from the literature and its contribution to the literature should be explained under this title.
4. Data and Methodology
Ø It should be explained why the renewable energy variable is chosen as a control variable.
Ø The rationale for the chosen method and tests should be explained.
5. Results and Discussion
Ø We should see the results of the Breusch – Pagan test and the Wald Test.
Ø It should be explained why the study was analyzed through four models.
Ø The significance levels in Table 2 should be explained with which values they are compared. Significance levels and critical values should be reported in the tables.
Ø Analyzes and the findings have not been adequately discussed.
Ø Different tests should be used as a method since the relationship and effect analysis is made between the variables. In addition, analyses should be made with new generation econometric tests. It is necessary to use up-to-date methods to publish articles in quality journals.
6. Conclusion
Ø The conclusion and recommendations section is very successful.
I consider it appropriate to MAJOR REVISION the study for the above-mentioned reasons. I would like to see the revised version of the paper before making the final decision.
Round 2
Reviewer 2 Report
This revised paper shows some improvement over the previous version of the manuscript. The authors have also addressed most of my concerns about the paper and have included some additional results of econometric tests in the Appendix. However, it requires a few minor revisions before considering for publication.
Reviewer 3 Report
Most of what was requested in the first report was not done. Therefore, I believe that the study does not have scientific competence. I recommend that you take the opinions of the referees into more consideration in your future studies.
